# Potential Chemotherapeutic Effect of Selenium for Improved Canceration of Esophageal Cancer

**DOI:** 10.3390/ijms23105509

**Published:** 2022-05-14

**Authors:** Anil Ahsan, Zhiwei Liu, Ruibing Su, Chencai Liu, Xiaoqi Liao, Min Su

**Affiliations:** Institute of Clinical Pathology, Guangodng Provincial Key Laboratory of Infectious Disease and Molecular Immunopathology, Shantou University Medical College, No. 22 Xinling Road, Shantou 515041, China; anilahsan@stu.edu.cn (A.A.); 12zwliu1@stu.edu.cn (Z.L.); leonsu@stu.edu.cn (R.S.); 20ccliu1@stu.edu.cn (C.L.); 13xqliao@stu.edu.cn (X.L.)

**Keywords:** selenium, esophageal squamous cell carcinoma, chronic inflammation, DNA damage

## Abstract

Esophageal squamous cell carcinoma is the most common type of esophageal cancer and accounts for 5% of malignant tumor deaths. Recent research suggests that chronic inflammation and DNA damage may drive the onset of esophageal squamous cell carcinoma, implying that lowering chronic inflammation and DNA damage compounds may provide chemo-prevention. According to epidemiological and experimental evidence, selenium is linked to a lower risk of several malignancies, including esophageal squamous cell carcinoma. However, its exact mechanism is still unclear. In the present study, we used cell lines and a 4-NQO mice model to explore the anti-cancer mechanism of four types of selenium. Our findings indicated that selenium inhibited the proliferation, colony formation, and ROS level of ESCC cell lines in a time-dependent manner. Intriguingly, selenium treatment impeded 4-NQO-induced high-grade intraepithelial neoplasia and reduced the number of positive inflammatory cells by preserving DNA from oxidative damage. In addition, selenium significantly decreased the expression of Ki-67 and induced apoptosis. This study demonstrates that selenium has a significant chemo-preventive effect on ESCC by reducing high-grade dysplasia to low-grade dysplasia. For the first time, selenium was shown to slow down the progression of esophageal cancer by lowering inflammation and oxidative DNA damage.

## 1. Introduction

Esophageal squamous cell carcinoma (ESCC) is the deadliest form of gastrointestinal malignancy and is the eighth most common cancer worldwide. It is the most prevalent in Asia and Africa, but North America and Europe are experiencing a rapid increase in cases [1,2]. This cancer is generally only diagnosed at later stages and has a poor 5-year survival rate due to limited treatment options [3]. The limited improvements in treatment outcomes provided by conventional therapies have prompted us to seek new drugs for treating ESCC that address the significant shortcomings of existing therapeutics.

Several previous studies indicate that inflammation accelerates cell cycle progression, cell proliferation, tumor vascularization, and the evasion of apoptotic cell death, all of which play a role in cancer initiation and progression [4]. Damage to critical biological components (e.g., DNA, proteins, and lipids) caused by chronic inflammation-induced reactive oxygen and nitrogen species (RONS) might lead to malignant cell transformation [5,6]. Our previous studies reported a significant increase in chronic-inflammation-related DNA damage responses as precancerous lesions progressed in three human gastric cardia tissues [7,8,9]. These results indicate that reducing inflammation and DNA damage may be a potential cancer-preventive and therapeutic technique.

Selenium (Se) is a vital and unique trace element that helps to prevent various tumor malignancies, including ESCC [10,11]. Previous studies described a significant inverse correlation between serum selenium levels and the risk of ESCC in China’s high-risk areas [12,13]. Many selenium compounds, including sodium selenite, selenate, methylselenic acid, methyl-selenocysteine, and Se-Met, were proven to be effective antitumorigenic agents in both basic research and clinical investigations [14,15]. The underlying anti-neoplastic mechanism of selenium has been described as anti-oxidation [16], anti-inflammatory [17], and causes apoptosis [18]. However, the precise role of selenium in ESCC is elusive. Therefore, additional studies of the potential interfere of selenium in ESCC will be valuable to illustrate its underlying biomarkers and provide therapeutic targets of ESCC.

## 2. Results

### 2.1. Selenium Inhibits the Proliferation and Colony Formation of ESCC Cell Lines

We first examined three established human ESCC cell lines, KYSE 150, KYSE 520, and CSEC 216, for their sensitivity to MSA, MSC, and Se-Met in comparison with sodium selenite using CCK8 cytotoxicity assays. The cells were treated with 1–100 µM selenite, 10–1000 µM MSC or Se-Met, and 0.001–6 µM MSA for 2 and 6 days. All three cell lines were susceptible to selenite, which induced a dose- and time-dependent growth inhibition (Figure 1A). At the same time, the significant effect of MSA was observed in ESCC cancer cells when its concentration was >1 µM (Figure 1B). However, the inhibitory effect of 2 days of treatment was less prominent for MSC and Se-Met than other Se compounds (Figure 1C,D). When each cell line was examined for sensitivity to MSC, Se-Met (100 µM) was able to significantly suppress the growth of cells with prolonged incubation (6 days). Because significant inhibition occurred in KYSE 150, KYSE 520, and CSEC 216 cells treated with 6 µM (selenite), 100 µM (MSC, Se-Met), and 2 µM (MSA), and higher doses of the respective selenium, these doses were used in each of the subsequent experiments.

The anti-proliferative capacity of selenium was also evaluated by colony formation assay. Three ESCC cell lines treated with 6 µM (selenite), 100 µM (MSC, Se-Met), and 2 µM (MSA) showed an inhibition of their colony-forming efficacy during 14 days of culture with a regular change of normal medium, every three days (Figure 1E–G), which was confirmed by a statistical analysis of colony-forming efficiency. These data clearly indicate that selenium compounds may possess a broad inhibitory effect on the growth of ESCC cells, despite having a different degree of anti-proliferation in different ESCC cell lines.

### 2.2. Selenium Controls the High-Grade Intraepithelial Dysplasia of Mice Induced by 4-NQO

The anti-cancer effect of selenium compounds was further investigated in C57BL/6 mice by using 4-Nitroquinoline 1-oxide (4-NQO), which is thought to be the most effective chemical for inducing in animals histological alterations similar to those seen in humans. A smooth surface, normal thickness, and flexibility were observed in the untreated esophageal tissue of mice. However, we observed obvious lesions in all 4-NQO-treated mice. The effects of selenium compounds were quite remarkable (Figure 2B). Upon gross examination, 4-NQO-treated mice developed both a high number (9.3 ± 1.19, *p* < 0.001) and large size of lesions (0.88 ± 0.22, *p* < 0.002), compared to untreated mice, respectively. Meanwhile, mice treated with selenium compounds had a significantly lower number of lesions: selenite group (0.05 mg/kg, 4.6 ± 1.13, *p* < 0.005), MSA group (0.05 mg/kg, 2.6 ± 1.13, *p* < 0.0001), MSC group (0.1 mg/kg, 4.2 ± 1.13, *p* < 0.002), Se-Met group (0.1 mg/kg, 2.7 ± 1.19, *p* < 0.0003; Figure 2C) and a significantly reduced size of lesions: selenite group (0.23 ± 0.14, *p* < 0.0005), MSA group (0.19 ± 0.16, *p* < 0.001), MSC group (0.28 ± 0.15, *p* < 0.002), Se-Met group (0.28 ± 0.17, *p* < 0.01; Figure 2D) in 4-NQO-treated mice, respectively. Additionally, the general status of the mice, which was assessed by monitoring their eating, drinking, and movements, was noticeably better in the mice treated with selenium compounds than those in the 4-NQO group.

Moreover, we observed microscopical morphological changes in esophageal tissues. We found that the esophageal epithelium of the 4-NQO-treated group exhibited high-grade intraepithelial neoplasia (HGIN) and more inflammatory cell infiltration than the untreated group. At the same time, selenium compounds (selenite, MSA, MSC, and Se-Met) showed low-grade intraepithelial neoplasia (LGIN). Nevertheless, the range of esophageal epithelial lesions in the selenite group was relatively closer to high-grade intraepithelial neoplasia. Similarly, the number of abnormal esophageal epithelial lesions in the MSA group was relatively lower, and inflammatory infiltration was also relatively lower (Figure 2E). This result suggests that it is likelier that selenium compounds prevent ESCC progression rather than reversing it entirely.

### 2.3. Effectiveness of Selenium for Preventing the Development of ESCC Lesions

To gain insights into the possible mechanisms underlying the inhibitory effects of selenium compounds in ESCC cell lines (KYSE 150, KYSE 520, and CSEC 216). The present study assessed their effects on apoptosis by flow cytometry. Annexin V-positive (early and late apoptosis) cells were considered the apoptotic population. ESCC cell lines were treated with selenium compounds—selenite (6 µM), MSA (2 µM), MSC (100 µM), and Se-Met (100 µM)—for 2 and 6 days. Our results indicate that all selenium compounds significantly increased cell apoptosis in KYSE 150 (Figure 3A,B), KYSE 520 (Appendix A), and CSEC 216 (Appendix A) compared with control in a time-dependent manner. Selenite showed larger apoptotic cells than other selenium compounds.

Next, we examined the morphology of three ESCC cell lines treated with selenium compounds for 2 and 6 days. The results show that the number of cells in an unscratched area reduced. The cells showed typical apoptotic morphology, including a detachment from the bottom of the culture dish and a change in cell shape (changed from an epithelial cell shape to a round-like cell shape) under a phase-contrast microscope (Figure 3C,D) and (Appendix A). Our results for the morphology examination of ESCC cell lines are consistent with the flow cytometry analysis.

We identified whether selenium compounds could also reduce proliferation and induce apoptosis in the esophageal epithelium tissue and investigated the proliferative status of these esophageal lesions after selenium treatment by detecting Ki-67 expression. A weak nuclear immunoexpression was observed in mice of the untreated group. When mice were exposed to 4-NQO, cell proliferation was highly elevated in the basal cells of the esophageal epithelium. In contrast, there was a significant decrease in Ki-67 expression in epithelial cells following selenium compound treatment (Figure 3E,F). To further investigate how selenium compounds inhibited esophageal epithelium proliferation, we evaluated apoptotic cells after the treatment of selenium compounds with TUNEL staining, which identified many apoptotic cells with bright green fluorescence. Our data indicated that apoptotic basal and mature suprabasal cells were frequently observed in esophagus sections of selenium-compounds-treated mice, but not in that of 4-NQO mice (Figure 3G,H). These results supported the effectiveness of selenium compounds for preventing the development of ESCC lesions, which might be achieved by the inhibition of cell proliferation and the induction of apoptosis in the esophageal epithelium.

### 2.4. Increased Expression of Inflammatory Markers in ESCC That Are Reversible in Treatment with Selenium

Chronic inflammation is linked to the development of cancer [19]. Our previous study indicated a link between chronic inflammation and DNA damage, as well as precancerous esophageal lesions [7,8,9]. To this end, we investigated whether inflammatory processes were involved in the molecular mechanisms of selenium against ESCC. We performed an immunofluorescence analysis for NF-κB (green fluorescence) on two human esophageal immortalized epithelial cell lines (NE2 and NE6) and quantified the ratio of nuclear to cytoplasmic NF-κB by an image analysis. NF-κB is a master regulator of the inflammatory process, responsible for the overall systemic inflammatory process [20]. Both cell lines (NE2 and NE6) treated with H_2_O_2_ increased nuclear fluorescence regarding the control. Thus, H_2_O_2_ seems to increase inflammation in both cell lines. In contrast, the combined exposure of H_2_O_2_ and selenium compounds markedly decreased nuclear fluorescence signals to those of only H_2_O_2_ treatment (Figure 4A–D), respectively.

Furthermore, we investigated whether selenium compounds showed anti-inflammatory effects in the 4-NQO mice model, and we performed an immunohistochemistry of mice esophagus sections to detect the CD45, CD4, and CD8 lymphocyte cells. The results showed that the numbers of positive lymphocyte cells (CD45, CD4, and CD8) per analyzed field were significantly higher (77.7 ± 8.6, *p* < 0.001, 30.2 ± 4.0, *p* < 0.001, 16.4 ± 1.8, *p* < 0.001, respectively) in the esophagus tissues of the 4-NQO group compared with those in the untreated group (Figure 4E–H). Nevertheless, the treatment with selenium compounds—selenite group (0.05 mg/kg), MSA group (0.05 mg/kg), MSC group (0.1 mg/kg), and Se-Met group (0.1 mg/kg)—significantly lowered the CD45 cells to 45.2 ± 5.7, *p* < 0.0001; 23.1 ± 9.1, *p* < 0.0001; 44.1 ± 7.1, *p* = 0.0001; and 31.1 ± 6.0, *p* < 0.0001 (Figure 4E,F), compared to 4-NQO-treated mice, respectively. In line with this, CD4-positive cells were reduced to 19.7 ± 2.9, *p* = 0.007; 15.1 ± 3.4, *p* = 0.0009; 18.3 ± 3.0, *p* = 0.003; and 14.6 ± 3.2, *p* = 0.0001 (Figure 4E,G) compared to mice given 4-NQO, respectively. Similarly, CD8-positive cells decreased to 6.4 ± 1.4, *p* < 0.0001; 3.0 ± 1.4, *p* < 0.0001; 5.1 ± 1.4, *p* < 0.0001; 5.2 ± 1.4, *p* < 0.0001 (Figure 4E,H) compared to 4-NQO-treated mice, respectively. Our findings suggest that selenium compounds are effective against ESCC, even in vitro model, and they might be regarded an effective anti-inflammatory molecule.

### 2.5. Selenium Treatment Results in Lower ROS Stress and Oxidative DNA Damage in ESCC

ROS-induced oxidative DNA damage is involved in precancerous esophageal lesions and tumor initiation. Consequently, excessive reactive oxygen species (ROS) accumulation plays a vital part in developing human esophageal cancer since ROS produces oxidative alterations of cellular macromolecules, such as DNA, proteins, and lipids [7,8]. We thus hypothesize that selenium might reduce ROS stress and oxidative DNA damage. To confirm the anti-oxidant effects of selenium, a fluorescent dye of DCFH-DA was used to measure the levels of ROS in three ESCC cell lines, which were treated with selenium compounds for 1 and 3 days. Treatment with selenite—(6 µM), MSA (2 µM), MSC (100 µM), and Se-Met (100 µM)—for 1 day significantly reduced ROS levels in ESCC cell lines. However, a prolonged treatment time for 3 days caused a further decrease in ROS levels. Selenium-reduced ROS levels were confirmed by comparing the fluorescence intensities of DCFH-DA-stained selenium-treated ESCC cell lines: KYSE 150 (Figure 5A,B), KYSE 520 (Appendix A), and CSEC 216 (Appendix A).

In addition, our data indicated that MSA showed a more anti-oxidant effect in KYSE 150 and KYSE 520, while MSC displayed more in CSEC 216 than other selenium compounds. γ-H2AX phosphorylation is a sensitive sign of DNA double-strand breaks, whereas 8-hydroxy deoxyguanosine is a hallmark of oxidative DNA damage. (DSB) [21,22]. Therefore, we determined DNA damage in selenium compounds treated in three ESCC cell lines after 1 and 3 days of incubation by the presence of the phosphorylated histone variant H2AX (γ-H2AX) and 8-hydroxydeoxyguanosine (8-OHdG) in the nucleus. Our results indicated that treatment with selenium compounds reduced the number of γ-H2AX foci after 1 day in three ESCC cell lines and prolonged incubation with selenium compounds for 3 days, causing a significant reduction in the number of γ-H2AX foci per cell as compared to the untreated cells: KYSE 150 (Figure 5C–E), KYSE 520 (Appendix A) and CSEC 216 (Appendix A). Consistently, the nuclear fluorescence intensity level of 8-OHdG was lowered in selenium compounds treated in three ESCC cell lines tested in comparison to untreated cells: KYSE 150 (Figure 5F–H), KYSE 520 (Appendix A), and CSEC 216 (Appendix A). To further support this notion, we also evaluated the DNA damage in two human esophageal immortalized epithelial cell lines (NE2 and NE6) after cells were pre-treated with selenium compounds for 2 h and then co-cultured with (800 µM) H_2_O_2_ for 16 h. The treatment of cells with H_2_O_2_ markedly increased the number of γ-H2AX foci (Appendix A) and nuclear fluorescence intensity level of 8-OHdG (Appendix A) relative to untreated control. Selenium compounds treatment significantly reversed the increase in the number of γ-H2AX foci and 8-OHdG intensities induced by H_2_O_2_. As noted in other studies, 8-OHdG staining also occurs outside the nucleus due to RNA and mitochondrial DNA.

We next tested the anti-oxidant effect of selenium compounds in the 4-NQO mice model through immunohistochemical investigations. We found that the immunoreactivity of γ-H2AX in the 4-NQO group was significantly higher than that in the untreated group (26.4 ± 1.7, *p* < 0.001). However, a decrease in γ-H2AX immunoreactivity was observed in mice that received selenium compounds: selenite group (0.05 mg/kg, 13.2 ± 1.7, *p* < 0.0001), MSA group (0.05 mg/kg, 8.6 ± 2.1, *p* < 0.0001), MSC group (0.1 mg/kg, 12.8 ± 1.7, *p* < 0.0001), Se-Met group (0.1 mg/kg, 9.7 ± 1.8, *p* < 0.0001) (Figure 5I,J) compared to 4-NQO-treated mice, respectively. These consistent results were observed for 8-OHdG between 4-NQO and selenium compounds: selenite group (80.6 ± 8.5, *p* < 0.0003), MSA group (59.7 ± 9.7, *p* < 0.0001), MSC group (63.4 ± 8.2, *p* < 0.0001), Se-Met group (68.3 ± 8.5, *p* < 0.0001), (Figure 5K,L) that of 4-NQO-treated mice, respectively. Collectively, our data indicated that anti-oxidant properties were shown by selenium compounds associated with preventing DNA from being oxidized.

## 3. Discussion

Esophageal cancer is one of the most common malignancies. ESCC occurs at very high rates in certain regions of China [3]. Currently, there are limited clinical approaches for the treatment of ESCC. Although significant progress in chemotherapy has been made in the past fifty years, the long-term survival for patients with ESCC has remained relatively unchanged. There are increasing shreds of evidence demonstrating that selenium could act as a potential anti-esophageal cancer agent [18], but its precise mechanisms are still not completely understood. Using ESCC cell lines, we found that selenium significantly reduced cell growth (Figure 1). Here, for the first time, to our knowledge, selenium’s systemic administration showed a prominent effect on reducing ESCC under our experimental state by showing low-grade intraepithelial neoplasia (Figure 2). Our further investigation found that selenium protected DNA against oxidative damage, both in vitro and in vivo, and thus protected against ESCC. These results suggest that selenium could be an excellent candidate for an esophagus cancer preventive agent. Hence, in the present study, we did not investigate the anti-cancer effect of selenium at initiation and different times of selenium doses at progression induced by 4-NQO, which should be addressed in further studies.

One of the physiological changes in cancer cells is their continued development paired with their loss of apoptotic mechanisms, which is associated with increased cell proliferation [23]. A significant finding from this study was that selenium remarkably reduced the growth of ESCC cell lines and the expression of Ki-67 in 4-NQO-treated mice, which indicated the anti-proliferative effects of selenium. Selenium’s anti-proliferative effects were also linked to apoptosis induction, as shown by increased numbers of apoptotic cells in three ESCC cell lines and increased numbers of TUNEL-positive cells in mice. These findings are consistent with earlier studies that display apoptosis and the anti-proliferative effects of selenium in different cell lines of tumors [24] and against oral carcinoma [25]. However, the apoptotic and anti-proliferative effects of selenium have not been reported in a 4-NQO mice model [26,27]. Methylseleninic acid confirmed the proliferation reduction and apoptosis induction in ESCC cell lines and the xenografts of treated mice [18,28,29]. Our study indicates that the inhibition of ESCC in mice receiving selenium was associated with a decreased cell proliferation and activation of apoptosis.

Inflammation is a natural aspect of the human body’s reaction to tissue damage when inflammation becomes persistent and is thought to be a significant risk factor for various human malignancies, including cancer [30]. Our previous findings reveal a link between chronic inflammation and precancerous esophageal lesions. In addition, selenium is an anti-oxidant that regulates thyroid function, improves male fertility, and has anti-inflammatory properties in various diseases [8,17]. Chronic inflammation induces early cancer changes by activating lymphocyte infiltration and improper pro-inflammatory mediator productions and transcription factors, such as the nuclear transcription factor kappa B (NF-κB) [31]. Accumulating evidence suggests that the NF-kB pathway plays an important role in the promoting phase of carcinogenesis by regulating immune function, inflammation, apoptosis, proliferation, stress responses, and the progression and invasion of cancer cells [32]. Interestingly, we found that selenium effectively reduced 4-NQO-induced esophagus inflammation by decreasing CD45, CD8, and CD4 T-cell markers in mice and the nuclear/cytoplasmic ratio of NF-κB in H_2_O_2_-treated human immortalized esophageal epithelial cell lines.

DNA damage can be caused by a variety of hazardous substances, including inflammatory cytokines [33]. Reactive oxygen species (ROS) and reactive nitrogen species (RNS) are produced in a chronic inflammatory condition and cause DNA damage. DNA double-strand breaks (DSBs) are the most severe type of DNA damage and are notoriously difficult to repair [7]. Accumulating evidence suggests a chemo-preventive role of selenium in cancer risk and incidence. Selenium functions as an anti-oxidant and plays a possible chemo-preventive role against cancer through scavenging reactive oxygen species (ROS), thereby preventing damage to the DNA and occurrence of mutations [34]. The effects of selenium status on cancer were studied in several clinical trials and epidemiologic studies in humans, suggesting the beneficial effects of higher selenium status in preventing the recurrence of lung cancer [35] and patients with different cancer types [11]. We thus hypothesized that selenium might reduce ROS production and DNA damage against ESSC. Interestingly, we found that selenium effectively reduced ROS levels in ESCC cell lines. Our further investigations suggested that selenium blocks both oxidative stress-induced guanine base damages (8-OHdG) and double-strand breaks (γ-H2AX) in both ESCC cell lines and 4-QNO treated mice. These results are consistent with our hypothesis. Thus, selenium could be a potential chemotherapeutic agent against the canceration of esophageal cancer.

## 4. Materials and Methods

### 4.1. Reagents and Antibodies

Sodium selenite (selenite), Se-(methyl) selenocysteine hydrochloride (MSC), selenomethionine (Se-Met), and methylseleninic acid (MSA) were purchased from Sigma (St. Louis, MO, USA). Dulbecco’s Modified Eagle Medium (DMEM), fetal bovine serum (FBS), trypsin–EDTA Roswell Park Memorial Institute (RPMI) 1640 Medium, keratinocyte–SFM medium, and an Epilife medium were acquired from Gibco (Grand Island, NY, USA). Penicillin and streptomycin were bought from Sigma (St. Louis, MO, USA). WST-8 [2-(2-methoxy-4-nitrophenyl)-3-(4-nitrophenyl)-5-(2,4-disulfophenyl)-2H-tetrazolium, monosodium salt] (CCK 8) was purchased from (Nanjing KeyGen Biotech. Co., Ltd., Nanjing, China). The carcinogen 4-NQO was purchased from Sigma Aldrich (St. Louis, MO, USA). Pentobarbital sodium was purchased from Sigma (St. Louis, MO, USA). Annexin V-FITC/PI Apoptosis Detection kit was purchased from (Vazyme Biotech Co., Ltd., Catalog # A211, Nanjing, Jiangsu, China). 2, 7-dichlorodihydrofluorescein diacetate (DCFH-DA) was obtained from (Solarbio, Beijing, China, Item No.: CA1410). Antifade Mounting with DAPI was obtained from Vector lab, Inc. (H-1200, Burlingame, CA, USA). Antibodies against Ki67 (#12202), γH2AX (#9718), NF-ĸB (#8242), CD45 (#70257), CD4 (#25229) and CD8 (#98941) were purchased from Cell Signaling Technology (Danvers, MA, USA). 8-OHdG (bs-1278R) was purchased from Bioss Biotechnologies, Beijing, China. Normal goat serum was bought from ZSGB-BIO, ZLI-9022, Beijing, China. Secondary antibodies (Max Vision HRP rabbit kit) were acquired from MXB Biotechnologies, Kit-5004, Fuzhou, China. Immunofluorescence secondary antibodies were obtained from (ab150077, Abcam, Cambridge, MA, USA). DAB Kit was obtained from MXB Biotechnologies, DAB-0031, Fuzhou, China).

### 4.2. ESCC Cell Lines and Culture Conditions

Three ESCC cell lines (KYESE 150, KYSE 520, and CSEC 216) and two human esophageal immortalized epithelial cell lines (NE2 and NE6) were included in this study. The ESCC cell lines were gifts from Shantou University Medical College [36]. CSEC 216 were routinely cultured in DMEM supplemented with 10% fetal bovine serum, 10 U/mL penicillin, and 10 U/mL streptomycin at 37 °C in a humidified atmosphere with 5% CO_2_. KYSE 150 and KYSE 520 were maintained in RPMI 1640 with 10% fetal bovine serum and antibiotic mixture at 37 °C in a humidified incubator containing 5% CO_2_. Cells were passaged upon reaching 70% to 80% confluency. Cells at passages 4–8 were used for all experiments.

*NE2 and NE6 cell culture:* Prof. George Tsao kindly provided the human immortalized esophageal epithelial cell lines (NE2 and NE6) from Hong Kong University. NE2 and NE6 cells were grown in a defined keratinocyte–SFM medium and an Epilife medium supplemented with a defined keratinocyte–SFM growth supplement. NE2 cells were accompanied with an additional 1% FBS. Cell cultures were maintained at 37 °C in a humidified atmosphere of 95% air and 5% CO_2_. The culture medium was renewed every three days. selenite, MSC, MSA, and Se-Met were prepared in double-distilled water, filter sterilized, and then stored in 100 µL aliquots and kept at −20 °C in a freezer. A fresh vial was thawed for each use. All experiments were performed with mycoplasma-free cells. All human cell lines were authenticated within the last 3 years using STR profiling.

### 4.3. Cell Counting Kit-8 (CCK8) Assay and Morphological Examination

Cell viability was estimated by CCK colorimetric assay and carried out as previously described [37]. See Appendix A for more details.

### 4.4. Clonogenic Assay

Single-cell suspensions were generated for each ESCC cell line, and a specified numbers of cells were seeded into six-well tissue culture plates. Then, cells were exposed to different doses of Se compounds (selenite 6 µM, MSA 2 µM, MSC 100 µM, and Se-Met 100 µM) for 2 weeks at 37 °C. Culture medium was replaced every three days until the appearance of colonies. The cells were fixed with 4% buffered formalin for 15 min and then stained with 1% crystal violet (Sigma Aldrich, St. Louis, MO, USA) for 30 min. The plates were gently washed with PBS. Cell clusters with >50 cells were considered as a colony, and colonies were imaged and counted by the software ImageJ.

### 4.5. Measurement of Apoptosis

Cell apoptosis was quantified using the Annexin V-FITC/PI Apoptosis Detection kit following the manufacturer’s protocol. Briefly, after indicated treatment, cells of each ESCC cell line were collected by centrifugation, washed twice with ice-cold PBS, centrifuged at 1000 rpm for 5 min, then gently resuspended in 500 µL of binding buffer. Then, 5 µL Annexin V-FITC and 5 µL PI were added to each sample. The cell samples were incubated in the dark for 15 min at room temperature, detected using flow cytometry (The BD Accuri™ C6 Plus Flow Cytometer, BD Biosciences), and analyzed using BD Accuri C6 plus software. The apoptotic rate was calculated as the percentage of early + late apoptotic cells.

### 4.6. Quantification of Intracellular ROS

The intracellular ROS level of three ESSC cell lines (KYSE 150, KYSE 520, and CSEC 216) was determined at culture days of 1 and 3 using a 2, 7-dichlorofluorescin-diacetate (DCFH-DA) based kit following the manufacturer’s protocol. See Appendix A for more details.

### 4.7. Immunofluorescence Staining

Immunofluorescence staining was performed against phospho-H2AX, 8-OHdG, to examine the effect of Se compounds on oxidative DNA damage in three ESSC cell lines (KYSE 150, KYSE 520. CSEC 216) using a method similar to that previously described [36]: see Appendix A for more details.

### 4.8. Animals, Study Design, and Endpoints

Six-week-old C57BL/6 mice were purchased from Beijing Vital River Laboratory Animal Technology Co., Ltd., Beijing, China (NO. SCXK, 2019-0001). Five mice were maintained in each plastic cage in an air-conditioned room with a 12 h light/dark cycle. The temperature and relative humidity were kept at 25 °C ± 2 °C and 50 ± 10%, respectively. All mice were fed with a basal diet (Rodent Chow Product, KeAoXie Li feeds Co., Ltd, Beijing, China) and allowed free access to deionized water. All the animal procedures and experiments conducted in this study were approved by the Animal Ethics Committee of Shantou University Medical College (Permit NO. SUMC2021-228), and all efforts were made to minimize suffering to the animals. We used 4-Nitroquinoline 1-oxide (4-NQO) to induce esophageal lesions. Stock solution (1 mg/mL) was prepared weekly in propylene glycol, diluted in drinking water to a working concentration of 100 μg/mL, and stored at 4 °C. Drinking water containing 4-NQO was freshly prepared every week using deionized water and administered to the mice in light-shielded water bottles [38,39,40]. The design for the experimental studies is schematically shown in Figure 2A. After 1 week of quarantine, C57BL/6 mice (*n* = 30 total) were randomized into six groups: Normal group (*n* = 5): fed with water containing propylene glycol from the 1st week to 16th week, then fed only normal water for further 4 weeks (shaded in grey); 4-NQO group (*n* = 5): fed with water containing 4-NQO (100 μg/mL) from the 1st to 16th week (shaded in black) and maintained with regular drinking water for further 4 weeks; selenite group (0.05 mg/kg, *n* = 5); MSA group (0.05 mg/kg, *n* = 5); MSC group (0.1 mg/kg, *n* = 5); and Se-Met group (0.1 mg/kg, *n* = 5): administered through force-feeding via intragastric intubation using 22-gauge needle size (Harvard Apparatus, Inc., Holliston, MA) [41] at a volume of 10 mL/kg weight every Monday, Wednesday, Friday for 4 weeks of post completion and 16 weeks of 4-NQO exposure (4-NQO → Se drugs; shaded in blue). All mice were monitored daily for general behavioral abnormalities, signs of toxicity, illness, or discomfort.

After completing the experimental period (20 weeks), all mice were euthanatized with a 1% aqueous pentobarbital sodium solution (i p). The whole esophagus and stomach were opened longitudinally, and esophagus lesions were performed under white light and carefully identified. Tissues were fixed in freshly made 4% paraformaldehyde overnight at 4 °C, embedded in paraffin, and sectioned into 4 µm sections. A histopathologic assessment of esophagus sections was performed to examine the effects of Se compounds on the incidence of esophagus lesions. In addition to histologic evaluation, esophagus sections were immunostained for the proliferation marker Ki-67, oxidative DNA damage marker 8-OHdg, γ-H2AX, inflammatory marker CD45, CD4, and CD8, and tunnel assay. The selected dose was based on a previous study that showed the inhibitory effects of selenium compounds when tested on various types of cancer in vivo [42,43,44].

### 4.9. Histopathological Analysis

Two certified pathologists conducted the histological determination of squamous neoplasia at the Shantou University Medical College on the sectioned tissue samples. The esophagus sections were de-paraffinized, rehydrated, stained with H & E for histopathology, and observed by a light microscope. The ESCC precursor lesions observed were classified into the following types: Squamous dysplasia—cytological atypia (nuclear atypia: enlargement, pleomorphism, hyperchromasia, loss of polarity, and overlapping) and architectural (abnormal epithelial maturation). Low grade—involvement of only the lower half of the epithelium, with only mild cytological atypia. High grade—involvement of more than half of the epithelium or severe cytological atypia (regardless of the extent of epithelial involvement) according to WHO 2019 classification and reclassified as the same lesions using the binary classification system [45]. Lesion volume was calculated according to the following formula: V (mm^3^) = (width^2^ × length)/2.

### 4.10. Terminal Deoxynucleotidyl Transferase-Mediated dUTP Nick End Labeling (TUNEL) Assay

Apoptosis was assayed by TUNEL staining using the Yeasen Biotech Kit according to the manufacturer’s instructions. Briefly, formalin-fixed, paraffin-embedded (FFPE) tissue on the slides was washed with PBS (pH 7.4) and permeabilized with 0.1% Triton X-100 for 2 min, followed by incubation with 50 μL of TUNEL reaction mixture in a humidified chamber at 37 °C for 1 h in the dark. Then, sections were counterstained with DAPI and observed under a fluorescence microscope (Zeiss, Jena, Germany). We calculated the percentage of TUNEL-positive cells by dividing the number of TUNEL-positive nuclei by the total number of DAPI-positive nuclei in defined fields and multiplied by 100.

### 4.11. Immunohistochemical Staining (IHC)

Immunostaining of esophagus sections for Ki-67, γ-H2AX, 8-OHdG, CD45, CD4, and CD8 was performed on formalin-fixed paraffin-embedded (FFPE) esophagus sections using the Envision technique, Dako Real EnVision Detection System, and Peroxidase/DAB+ (Agilent Technologies, Santa Clara, CA, USA) according to the manufacturer’s protocol. See Appendix A for more details.

### 4.12. Data and Statistical Analysis

Statistical significance in the experiments was assessed using GraphPad Prism, version 8.02 (La Jolla, CA, USA). All data were collected and analyzed, blinded, and presented as mean ± SD. Statistical evaluations of the post hoc multiple group comparisons were conducted using one-way ANOVA and two-way ANOVA (analysis of variance), followed by Tukey contrasts. A *p* value ≤ 0.05 was considered statistically significant; NS, non-significant.

## 5. Conclusions

Based on our findings, we suggest that selenium possesses a robust chemotherapeutic effect against ESCC. This effect is related to promoting apoptosis, inhibiting the proliferation of cells, preventing oxidative stress, and reducing inflammatory T cells (CD45, CD4, CD8). Selenium could diminish high-grade dysplasia to low-grade dysplasia via multi-targeted effects, including anti-oxidants and anti-inflammatory effects (Figure 6). These data will potentially reduce the progression of esophageal squamous carcinoma and provide room for further studies on the fundamental, clinically applied levels and pharmacokinetic studies in selenium. There are some limitations to this study that should be acknowledged. In the present study, we could not identify the anti-cancer effect of selenium at initiation and different times of selenium doses at progression induced by 4-NQO, which would be beneficial and should be investigated in future studies. Furthermore, we did not elucidate the exact molecular pathway involved in the chemotherapeutic activity of selenium in ESCC; this would also be required in further investigations.

## Figures and Tables

**Figure 1 ijms-23-05509-f001:**
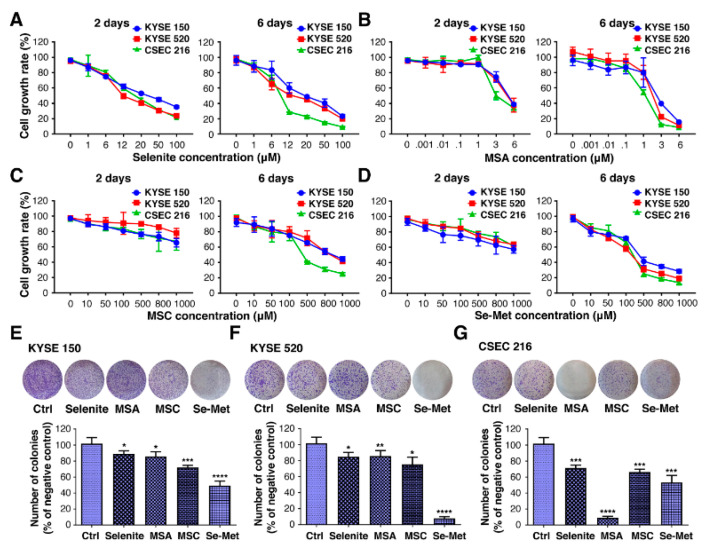
Selenium prevents cell proliferation and colony formation in ESCC cell lines. Respective ESCC cell lines were treated with increasing concentrations of (**A**) selenite, (**B**) MSA, (**C**) MSC, and (**D**) Se-Met for 2 days and 6 days, followed by the CCK8 assay and was calculated by the following formula: cell viability (%) = (OD_570_ of the treatment samples/OD_570_ of the control samples) × 100%. Results in triplicate represent the percentages (mean values ± SD) of cell growth to the untreated control. Colony formation assay was performed in (**E**) KYSE 150, (**F**) KYSE 520, and (**G**) CSEC 216 exposed to Se compounds for 14 days before being stained with crystal violet. Concentrations used were: 6 µM (selenite), 100 µM (MSC, Se-Met) and 2 µM (MSA). The cell colonies were photographed, and the number of colonies was counted. The differences were significant at (* *p* < 0.05, ** *p* < 0.01, *** *p* < 0.001, **** *p* < 0.0001 vs. untreated control, *n* ≥ 3). Data are expressed as mean ± SD from three (*n* = 3) independent experiments. The difference with *p* ≤ 0.05 was considered statistically significant.

**Figure 2 ijms-23-05509-f002:**
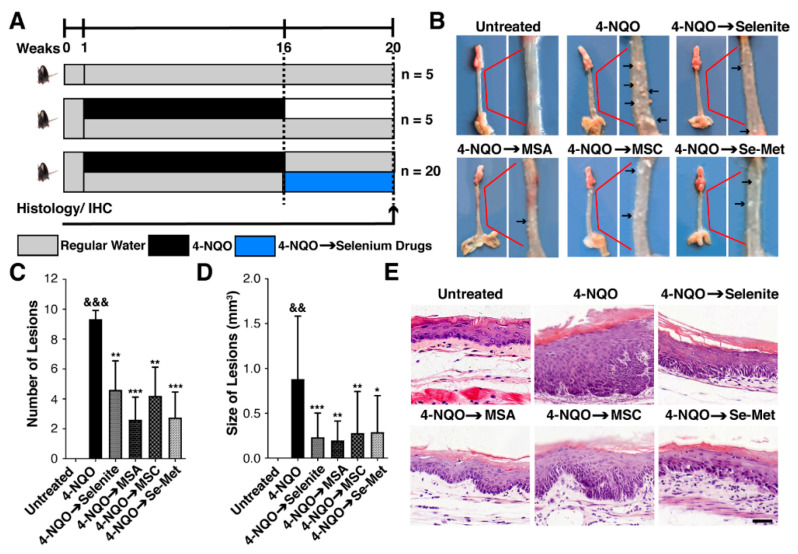
Selenium impedes ESCC lesions in the 4-NQO mice model. (**A**) Scheme for the treatment paradigm. C57BL/6 mice were randomized into six groups (*n* = 5, per group) and fed with water or 4-NQO (100 μg/mL) in drinking water for 16 weeks, after which all mice were given deionized water until week 20. Selenite (0.05 mg/kg), MSA (0.05 mg/kg), MSC (0.1 mg/kg), Se-Met (0.1 mg/kg) were administered via intragastric intubation on Monday, Wednesday, Friday after 16 weeks of 4-NQO exposure for 4 weeks. (4-NQO → Se compounds; shaded in blue). The histopathological assessment and immunohistochemistry of esophageal tissues were performed after 20 weeks: (**B**) Representative images of esophageal lesions for each treatment group. (**C**) Columns represent the number of esophageal lesions for each treatment group. (**D**) Columns represent the size of esophageal lesions for each treatment group. (**E**) Histological analysis of mice’s normal esophagus and ESCC esophagus for each treatment group (magnification, x400; scale bar = 40 μm). Differences are significant at (^&&&^ *p* < 0.001, ^&&^
*p* < 0.01 vs. untreated group, * *p* < 0.05, ** *p* < 0.01, *** *p* < 0.001 vs. 4-NQO group). The difference with *p* ≤ 0.05 was considered statistically significant.

**Figure 3 ijms-23-05509-f003:**
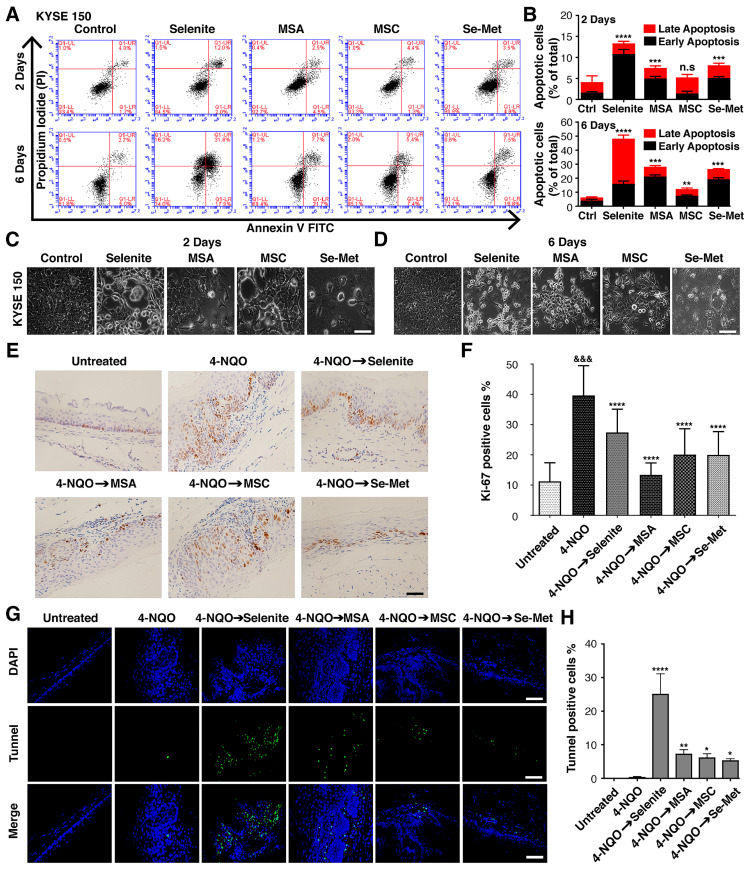
Effects of selenium on cell proliferation and apoptotic cell death against ESCC. Cells were treated with selenite (6 µM), MSA (2 µM), MSC (100 µM), and Se-Met (100 µM) for 2 and 6 days. Cell apoptosis was measured by flow cytometry. (**A**) Flow cytometry dot plot figures of apoptotic cells in KYSE 150. (**B**) Apoptotic cells represent the percentage of Annexin-V-single-positive and Annexin-V/PI-double-positive cells after 2 and 6 days in KYSE 150. In each dot plot figure, the upper left quadrant resembles necrotic cells; the upper right quadrant contains the later apoptotic cells, which are positive for Annexin V and propidium iodide (PI); the lower left quadrant shows viable cells, which exclude PI and Annexin V; the lower right quadrant denotes early apoptotic cells: Annexin-V-positive and PI-negative. Phase-contrast photomicrograph depicted representative morphological responses of KYSE 150 cell line at (**C**) 2 days and (**D**) 6 days. Differences are significant at (** *p* < 0.01, *** *p* < 0.001, **** *p* < 0.0001 vs. untreated control; n.s, non-statistically significant, *n* ≥ 3). Data are expressed as mean ± SD from three (*n* = 3) independent experiments. Scale bar, 20 μm. Selenium compounds, selenite (0.05 mg/kg), MSA (0.05 mg/kg), MSC (0.1 mg/kg), Se-Met (0.1 mg/kg) were administered for 4 weeks of post-completion and 16 weeks of 4-NQO exposure. The mice were sacrificed, and the esophagus sections were fixed, embedded, sectioned, and stained with anti-Ki-67 antibody and TUNNEL. Four to five representative areas of each esophagus section from each mouse were photographed and analyzed per group. (**E**) Immunohistochemical staining of Ki-67 for each treatment group. (**F**) Quantitative analysis of Ki-67-positive cells for each treatment group. (**G**) Fluorescence microscopy images of TUNEL and DAPI-stained nuclei (magnification, ×400; scale bar = 40 μm). (**H**) Quantitative analysis of selenium-induced apoptosis for each treatment group. Differences are significant at (^&&&^ *p* < 0.001 vs. untreated group, * *p* < 0.05, ** *p* < 0.01, **** *p* < 0.0001 vs. 4-NQO group). The difference with *p* ≤ 0.05 was considered statistically significant.

**Figure 4 ijms-23-05509-f004:**
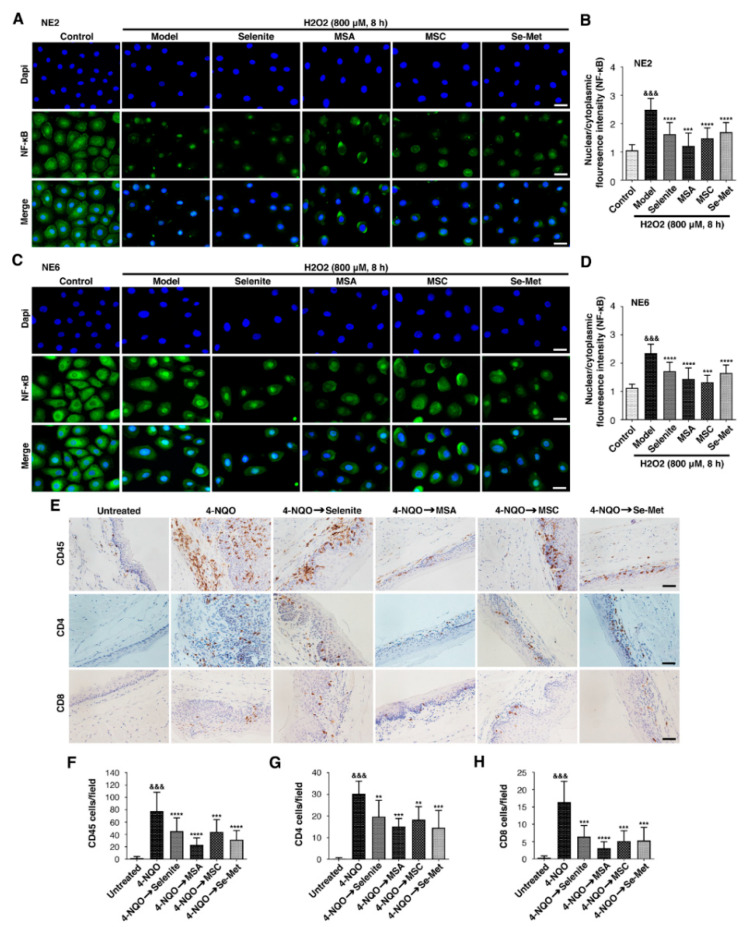
Selenium impedes inflammation in esophageal immortalized epithelial cell lines and 4-NQO mice model. Cells (NE2 and NE6) were pre-treated with selenium compounds, selenite (6 µM), MSA (2 µM), MSC (100 µM), and Se-Met (100 µM) for 2 h and then co-cultured with (800 µM) H_2_O_2_ for 8 h. NF-κB expression was visualized by immunofluorescence using primary specific antibodies and Alexa Fluor 488-conjugated secondary antibodies. Nuclei were stained with DAPI. (**A**) Representative images of NF-κB fluorescence in NE2 cell line. (**B**) The image analysis of NF-κB fluorescence data shows the nuclear to cytoplasmic staining ratio for each treatment in the NE2 cell line. (**C**) Representative images of NF-κB fluorescence in NE6 cell line. (**D**) The image analysis of NF-κB fluorescence data shows the nuclear to cytoplasmic staining ratio for each treatment in the NE6 cell line. The nuclear/cytoplasm fluorescence intensity per cell was determined by counting at least 100 cells on 10 fields randomly selected for each sample. Differences are significant at (^&&&^
*p* < 0.001 vs. untreated group, *** *p* < 0.001, **** *p* < 0.0001 vs. H_2_O_2_ group, *n* ≥ 3). Data are expressed as mean ± SD from three (*n* = 3) independent experiments. (Magnification, ×40; scale bar = 20 μm.) Selenium compounds, selenite (0.05 mg/kg), MSA (0.05 mg/kg), MSC (0.1 mg/kg), Se-Met (0.1 mg/kg) were administered for 4 weeks of post-completion and for 16 weeks of 4-NQO exposure. The mice were sacrificed, and the esophagus sections were fixed, embedded in paraffin, and sectioned. Then, the tissue sections were stained with CD45, CD8, and CD4 antibodies. Four to five representative areas of each esophagus section from each mouse were photographed and analyzed per group. (**E**) Representative immunohistochemical staining of CD45, CD4, and CD8 for each treatment group. (**F**) Numerical analysis of CD45-positive cells for each treatment group. (**G**) Numerical analysis of CD4-positive cells for each treatment group. (**H**) Numerical analysis of CD8 for each treatment group. Differences are significant at (^&&&^ *p* < 0.001 vs. untreated group, ** *p* < 0.01, *** *p* < 0.001, **** *p* < 0.0001 vs. 4-NQO group). A difference with p ≤ 0.05 was considered statistically significant. (Magnification, ×400; scale bar = 40 μm.).

**Figure 5 ijms-23-05509-f005:**
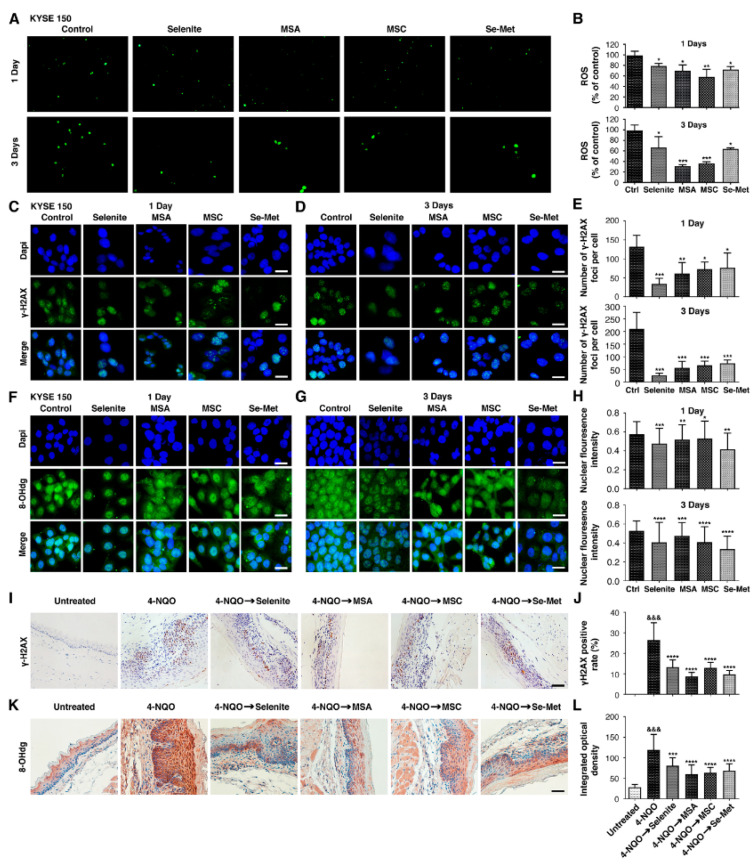
Selenium diminishes intracellular ROS levels and oxidative DNA damage in ESCC. Cells were treated with selenite—(6 µM), MSA (2 µM), MSC (100 µM), and Se-Met (100 µM)—for 1 and 3 days. (**A**) The level of intracellular ROS in KYSE 150 cells was stained by DCFH-DA and observed under Zeiss Vert A1 fluorescence microscope. (**B**) Statistical results were read by fluorescent plate reader in KYSE 150 cells. γ-H2AX expression and 8-OHdg expression were visualized by immunofluorescence using primary specific antibodies and Alexa Fluor 488-conjugated secondary antibodies. Nuclei were stained with DAPI. (**C**,**D**) Representative images of γ H2AX immunostaining after 1 day and 3 days in KYSE 150. (**E**) Columns represent the number of γ-H2AX foci/cell for each treatment after 1 day and 3 days in KYSE 150. (**F**,**G**) Representative images of 8-OHdg immunostaining after 1 day and 3 days in KYSE 150. (**H**). Histograms show nuclear fluorescence intensity signal quantification in the nuclei of each treatment after 1 day and 3 days in KYSE 150. Differences are significant at (* *p* < 0.05, ** *p* < 0.01, *** *p* < 0.001, **** *p* < 0.0001 vs. untreated control, *n* ≥ 3). Data are expressed as mean ± SD from three (*n* = 3) independent experiments. (Magnification, ×40; scale bar = 20 μm.) Selenium compounds, selenite (0.05 mg/kg), MSA (0.05 mg/kg), MSC (0.1 mg/kg), and Se-Met (0.1 mg/kg) were administered for 4 weeks of post-completion and 16 weeks for 4-NQO exposure. The mice were sacrificed, and the esophagus sections were fixed, embedded in paraffin, and sectioned. Then, the tissue sections were stained with γ-H2AX and 8-OHdg antibodies. Four to five representative areas of each esophagus section from each mouse per group were photographed and analyzed. (**I**) Immunohistochemical staining of γ-H2AX for each treatment group. (**J**) Statistical analysis of γ-H2AX-positive cells for each treatment group. (**K**) Fluorescence microscopy images of 8-OHdg immunostaining for each treatment group. (**L**) Statistical analysis of IOD of 8-OHdg immunostaining for each treatment group. Differences are significant at (^&&&^ *p* < 0.001 vs. untreated group, *** *p* < 0.001, **** *p* < 0.0001 vs. 4-NQO group). The difference with *p* ≤ 0.05 was considered statistically significant. (Magnification, ×400; scale bar = 40 μm.).

**Figure 6 ijms-23-05509-f006:**
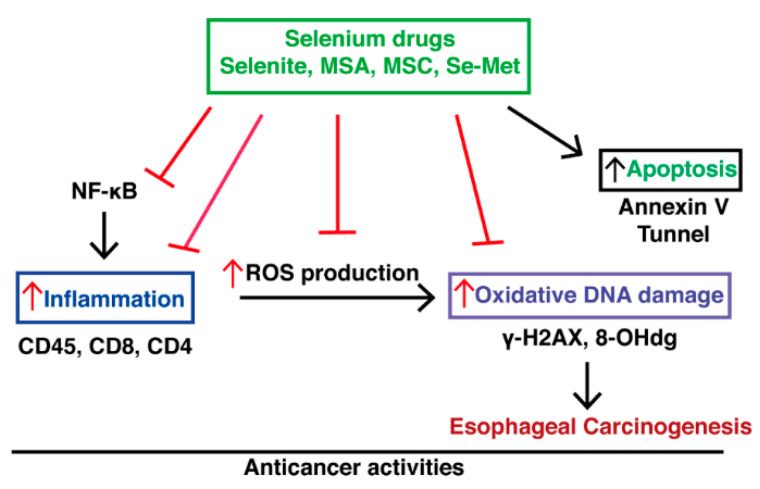
A detail mechanistic illustration showing the action of selenium against ESCC.

## Data Availability

The data that support the findings of our study are available from the corresponding author upon reasonable request.

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
