# Peer review of "Potential Chemotherapeutic Effect of Selenium for Improved Canceration of Esophageal Cancer"

_ijms, 2022, doi:10.3390/ijms23105509_

Round 1

Reviewer 1 Report

In the manuscript " Preclinical Evaluation of Selenium Halts the Canceration of Esophageal Cancer", the authors present a clinical study of the impact of Selenium in the evolution of esophageal cancer. The paper could have a clinical significance and an impact in the clinical practice. However, minor criticisms are present, as follows:

- The title should be changed as it does not have a great impact ;

- could the authors please explain in the results part why did they chose 2 and 6 days of selenium treatment

- The limitations of the study are not presented;

Author Response

Dear Prof. GAO,

Subject: Submission of the revised paper “Potential Chemotherapeutic Effect of Selenium for Improved Canceration of Esophageal Cancer” (ijms-1717762).

Enclosed are our responses to the comments on our manuscript (ijms-1717762) entitled Potential Chemotherapeutic Effect of Selenium for Improved Canceration of Esophageal Cancer. We highly appreciate all comments and revision suggestions. We respect the editorial/reviewer’s valuable comments and constructive suggestions for further improvement of this manuscript. We have made minor revisions following the reviewers’ comments in the revised manuscript.

Our point-to-point responses and corrections are displayed in red font in the following text. We have made careful revisions closely following the comments. Thus, the clarity and quality of our manuscript is improved. We sincerely hope that our response is satisfactory.

If you have any question, please let us know.

Sincerely,

Min Su

Institute of Clinical Pathology, Department of Pathology, Shantou University Medical College, No.22 Xinling Road, Shantou, Guangdong, 515031, PR China.

Prof. Min Su, Ph.D., (Corresponding authors)

Point by point list of responses to comments:

Response to Reviewer 1:

Point 1: The title should be changed as it does not have a great impact.

Response 1: Respected reviewer, we have changed the title as required in the revised manuscript.

Point 2: Could the authors please explain in the results part why did they chose 2 and 6 days of selenium treatment. 

Response 2: Thanks for this valuable comment. We chose 2 and 6 days of selenium treatment as organic Se compounds, such as MSA, Se-Met, and MSC, are less toxic than selenite. Methylselenol is the common active metabolite that is metabolized through hydrogen selenide. Organic Se compounds require enzymatic conversion to methylselenol produced by methioninase or b-lyase [1, 2]. Because the level of methioninase in mammalian cells is negligible, and cells in culture have low levels of b-lyase, it leads to the inefficient conversion of Organic Se compounds to methylselenol and delayed action [3].

  1. Zhao, R., F.E. Domann, and W. Zhong, Apoptosis induced by selenomethionine and methioninase is superoxide mediated and p53 dependent in human prostate cancer cells. Mol Cancer Ther, 2006. 5(12): p. 3275-84.
  2. Ip, C., Y. Dong, and H.E. Ganther, New concepts in selenium chemoprevention. Cancer Metastasis Rev, 2002. 21(3-4): p. 281-9.
  3. Ip, C., et al., In vitro and in vivo studies of methylseleninic acid: evidence that a monomethylated selenium metabolite is critical for cancer chemoprevention. Cancer Res, 2000. 60(11): p. 2882-6.

Point 3: The limitations of the study are not presented. 

Response 3: Thank you for your comment; according to your suggestion to strengthen the manuscript's current statement, we have added the limitation and future prospects of study in the revised submission.

Reviewer 2 Report

The manuscript entitled “Preclinical Evaluation of Selenium Halts the Canceration of Esophageal Cancer” by Ahsan et al. The idea summarized in the title is exciting, this research work needs typographical corrections and other minor corrections which have been highlighted in the manuscript. In my opinion this research paper will proves to be a much cited paper. I recommend to accept the paper with minor revision.

Please use the attached file for comments

Author Response

Dear Prof. GAO,

Subject: Submission of the revised paper “Potential Chemotherapeutic Effect of Selenium for Improved Canceration of Esophageal Cancer” (ijms-1717762).

Enclosed are our responses to the comments on our manuscript (ijms-1717762) entitled Potential Chemotherapeutic Effect of Selenium for Improved Canceration of Esophageal Cancer. We highly appreciate all comments and revision suggestions. We respect the editorial/reviewer’s valuable comments and constructive suggestions for further improvement of this manuscript. We have made minor revisions following the reviewers’ comments in the revised manuscript.

Our point-to-point responses and corrections are displayed in red font in the following text. We have made careful revisions closely following the comments. Thus, the clarity and quality of our manuscript is improved. We sincerely hope that our response is satisfactory.

If you have any question, please let us know.

Sincerely,

Min Su

Institute of Clinical Pathology, Department of Pathology, Shantou University Medical College, No.22 Xinling Road, Shantou, Guangdong, 515031, PR China.

Prof. Min Su, Ph.D., (Corresponding authors)

Point by point list of responses to comments:

Response to Reviewer 2:

Point 1: It is not clear which tests have been used. The authors are advised to name the statistical tests used for the current study.

Authors should also use an alternative post hoc test, such as LSD or Tukey's for multiple comparison. please revise the statistical analysis.

Response 1: Thank you for pointing out the missing ‘‘multiple comparisons’’ in Data and Statistical Analysis. Under your guideline, we have done and written the Tukey's for multiple comparisons in the revised manuscript.

Point 2: Discussion should be elaborated more as per the result sub-headings and should compare with the previous findings. I would recommend to provide the mechanistic approach how selenium halts the esophageal cancer.

Response 2: We highly appreciate the reviewer’s suggestion. We have revised our discussion by elaborating more as per the result sub-headings and further highlighting the mechanistic approach of the present study

Point 3: Please provide the detailed mechanistic picture.

Response 3: As required, we have provided a detailed mechanistic picture in the revised submission.

Point 4: Authors are advised to include the limitations and future prospects of the current study.

Response 4: Thank you for your comment; according to your suggestion to strengthen the manuscript's current statement, we have added the limitation and future prospects of study in the revised submission.
